# Kinetics Study of Polypropylene Pyrolysis by Non-Isothermal Thermogravimetric Analysis

**DOI:** 10.3390/ma16020584

**Published:** 2023-01-06

**Authors:** Ibrahim Dubdub

**Affiliations:** Department of Chemical Engineering, King Faisal University, Al-Ahsa 31982, Saudi Arabia; idubdub@kfu.edu.sa

**Keywords:** pyrolysis, polypropylene polymer, activation energy, thermogravimetric analyzer (TGA), kinetics

## Abstract

Polypropylene (PP) is considered as one of six polymers representative of plastic wastes. This paper attempts to obtain information on PP polymer pyrolysis kinetics with the help of thermogravimetric analysis (TGA). TGA is used to measure the weight of the sample with temperature increases at different heating rates—5, 10, 20, 30, and 40 K min^−1^—in inert nitrogen. The pyrolytic kinetics have been analyzed by four model-free methods—Friedman (FR), Flynn–Wall–Qzawa (FWO), Kissinger–Akahira–Sunose (KAS) and Starnik (STK)—and by two model-fitting methods—Coats–Redfern (CR) and Criado methods. The values of activation energies of PP polymer pyrolysis at different conversions are in good agreement with the average of (141, 112, 106, 108 kJ mol^−1^) for FR, FWO, KAS and STK, respectively. Criado methods have been implemented with the CR method to obtain the reaction mechanism model. As per Criado’s method, the most controlling reaction mechanism has been identified as the geometrical contraction models—cylinder model.

## 1. Introduction

Many researchers are focused on finding suitable ways to deal with huge plastic wastes in order to recycle such vast wastes. In our previous publication (Dubdub and Al-Yaari (2020, 2021) [1,2]), the problem of growing plastic wastes throughout the world has been outlined, as well as how it can be sorted by a “primary”, “secondary”, or “tertiary” recycling method with their limitations. The last option, “tertiary”, usually needs some advanced investigations, and one of the available methods is the “pyrolysis” method. This method has some advantages compared with the rest of the methods such as the combustion by the reduction in the volume of gaseous products (Kaminsky et al. (1996) [3]).

In this research direction, there is a limited amount of reported literature on PP as one of the major plastic waste constituents and their pyrolysis kinetics. Wu et al. (1993) [4] applied the pyrolysis for six polymers mixtures—high-density polyethylene (HDPE), low-density polyethylene (LDPE), PP, polystyrene (PS), Polyvinyl chloride (PVC), and acrylonitrile-butadiene-styrene (ABS) of municipal solid waste (MSW)—by TGA at three sets of heating rates (1, 2, and 5.5 K min^−1^). They found that the interaction between these polymers of MSW in the pyrolysis was insignificant. Diaz Silvarrey and Phan (2016) [5] studied the kinetic cracking of different polymers—PP, PS, HDPE, LDPE, and Polyethylene terephthalate (PET)—for TGA. They used Malek, KAS, and linear model fitting methods to obtain the pyrolysis mechanism. They ran the TGA with four heating rates (5, 10, 20, and 40 K min^−1^) and a temperature range of 30–700 °C. Their results showed that all the polymers followed the same one-step decomposition mechanism with the increase in temperature, and their order of decomposition was: PS ˂ PET ˂ PP ˂ LDPE ˂ HDPE. They also reported that the values of the activation energy and pre-exponential factor by the KAS method for PP are 261.22 KJ mol^−1^ and 3.03 × 10^21^ S^−1^. Chowlu et al. (2009) [6] studied the pyrolysis behavior for the mixture of two polymers, PP and LDPE, with five different mixture compositions and heating rates. They used the Vyazovkin (VYK) method (model-free technique) to evaluate the change in activation energy with the conversion. They found that this change in the reaction occurs in three different zones: slow at low conversion, slightly high at the middle conversion, and strongly high until the end of the decomposition. They concluded that the best mixture of the PP/LDPE weight ratio is 65%/35% since it has a low activation energy. Aboulkas et al. (2010) [7] calculated the activation energy and the reaction model of the pyrolytic reaction of HDPE, LDPE, and PP from non-isothermal TGA data. They obtained the values of activation energy by FR, KAS, and FWO between 179 and 188 kJ mol^−1^ for PP. They found the “geometrical contraction models—cylinder” to be the conversion model using the CR and Criado methods. Yu et al. (2016) [8] reviewed and compiled the various reported results on the pyrolysis of PVC mixed with PP/PE/PS. It has been reported that the interaction between the polymers depends mainly on the nature of the polymer itself. They also studied the effect of this different mixture of PVC and one of these polymers (PP/PE/PS) with the respect to: the onset temperature, maximum peak decomposition, residue weight, and quantity. Anene et al. (2018) [9] researched the cracking of LDPE/PP mixtures with different compositions in order to investigate the effect of the addition of PP to LDPE. They found that the cracking starts with a lower temperature for LDPE/PP compared to LDPE alone, informing the interaction between the two polymers. Finally, Mumbach et al. (2019) [10] performed the decomposition of plastic solid waste (PSW) by TG under inert conditions from 25 to 1000 °C with four heating rates (5, 10, 20, and 30 K min^−1^). They estimated that the feedstock of PSW will include: 17.28% LDPE, 7.41% HDPE, 51.85% PP, 17.28% plastics for PET, PS, and PVC, and 6.18% lignocellulosic materials. They evaluated the activation energy using KAS, STK, FWO, VYA, and STK (isoconversional methods) and the frequency factor by the compensation effect factor, while they evaluated the reaction model by the master plot. They identified that three main stages of reaction occur: the first decomposition occurs by the main decomposition of holocellulose and the minor decomposition of the first degradation stage of PVC (dichlorination); the second decomposition occurs mainly by a decomposition of a mixture of polymers, such as PS and some adhesive acrylic-based resins and PVC (dichlorination); and the third decomposition occurs by PP, HDPE, and LDPE as the major fraction and the second PVC thermal decomposition as the minor fraction. Galiwango and Gabbar (2022) [11] studied the co-pyrolysis of the PP polymer and paper wastes. They found that the activation energy values for the PP–paper mixture are much lower than those of pure PP, and this is because of the synergistic interactions between PP and paper wastes.

The objective of this study is to obtain the kinetic parameters of PP polymer pyrolysis using TGA data. Since the kinetic parameters are required for any further study, such as setting any type of reactor, the activation energy has been calculated using six methods (four model-free methods and two model-fitting methods). Dubdub and Al-Yaari (2020, 2021) [1,2] started a comprehensive study of the calculation of kinetics parameters for the pyrolysis of different polymers using TGA.

## 2. Materials and Methods

### 2.1. Materials and Thermogravimetry of PP

PP supplied from Ipoh SY Recycle Plastic/Malaysia has been used. PP pellets were ground into powder before putting them in the thermogravimetric analyzer. The pyrolysis of PP was performed by a 1020 series thermogravimetric TGA-7, manufactured by PerkinElmer Co., Waltham, MA, USA. A total of 10 mg of PP powder samples was used throughout all the TGA experiments in an inert atmosphere of pure N_2_ (99.999%) gas flowing at 100 cm^3^/min at five different heating rates (5, 10, 20, 30, and 40 K min^−1^). Each experiment was run more than once to ensure the reproducibility of the collected data. The international confederation for thermal analysis and calorimetry (ICTAC) recommends a ratio between the lowest and highest non-isothermal heating rates within 10–15 (Osman et al. (2022) [12]). In this work, a ratio of 8 (lowest = 5, highest = 40) is close to the minimum limit of 10. Both of the analysis results (ultimate and proximate) are presented in Table 1.

The 1020 series Thermogravimetric Analysis TGA7 has been employed to achieve the objective of this work. It has the following components: the controller with thermal analysis software and the thermogravimetric analyzer.

### 2.2. Kinetic Theory

There are two ways of running a TGA instrument: isothermal and non-isothermal. In this work, non-isothermal TGA is used over isothermal TGA because the non-isothermal TGA needs less time, and the isothermal TGA is not convenient at higher temperatures for large non-isothermal heat-up and cool-down times (Nawaz and Kumar (2022) [13]). During the pyrolysis, polymer chains break, producing volatile products and releasing them from the original polymer, which causes a loss of mass. Therefore, TGA is suitable for obtaining the kinetic parameters of the polymer (Hayoune et al. (2022) [14]). The estimation of kinetic variable with the data collected from the TGA can be performed by different methods. Those methods include one single TGA data (called the “model-fitting” method) or multiple TGA data at different heating rates (called the “model free” or “isoconversional” methods) and calculating the kinetic equation with differential methods or integrals.

The reaction kinetics of the PP pyrolysis can be started with the following equation:(1)$$\frac{d\alpha }{dt}=A\exp\left(-\frac{E_{a}}{RT}\;\right)f\left(\alpha \right)$$
where *α*, *t*, *E_a_*, *R*, *A*_0_, and *T* stand for the reaction conversion, the time, the activation energy, the universal gas constant, the frequency factor, and the absolute temperature, respectively.

For non-isothermal pyrolysis, *β* (heating rate) can be inserted into Equation (1):(2)$$\beta \frac{d\alpha }{dT}=A\exp\left(-\frac{E_{a}}{RT}\;\right)f\left(\alpha \right)$$

The derivation of the model-free methods (FR, FWO, KAS, and STK) and model-fitting methods, starting from Equation (2) with all the assumptions until the final equation, can be found in Aboulkas et al. (2010) [7]. Table 2 and Table 3 present the kinetic equations of the four commonly used model-free methods and the two model-fitting methods.

The Criado equation connects the reduced theoretical curve representing the characteristic of each reaction mechanism (left side) and the experimental data (right side). Therefore, a comparison between these two sides will inform which exact kinetic model will describe the experimental reaction. Table 4 shows the common solid-state thermal reaction mechanisms—*f*(*α*) and *g*(*α*)—used in the CR method and the Criado method.

## 3. Results and Discussion

### 3.1. Thermogravimetry of PP

The TG and the DTG with the set of heating rates of PP polymer pyrolysis are shown in Figure 1 and Figure 2, respectively. The thermograms were identical, and the thermal decomposition characteristics (onset, peak, and final temperatures) shown in Table 5 were shifted to a higher temperature with a higher heating rate; this shifting was also reported by different researchers (Wu et al. (1993) [4], Yang et al. (2001) [15], Park et al. (2000) [16]).

These figures (Figure 1 and Figure 2) prove only one reaction region for the pyrolysis of the PP polymer. This finding is in full agreement with different published data (Aboulkas et al. (2010) [7], Yu et al. (2016) [8], Anene et al. (2018) [9]).

There are two ways to calculate the kinetics parameters from TGA data: either a single thermogram or multiple thermograms at a set of heating rates, differentially or integrally (Chan and Balke (1997) [17]. In this paper, six methods have been implemented and compared between them. One set includes four model-free methods (FR, FWO, KAS, and STK), while the second set, model-fitting, implies one single thermogram (CR and Criado equation). The Criado equation method is considered the “masterplot” since it determines the kinetic model of the process.

### 3.2. Model-Free Kinetics Calculation

Model-free methods are preferable to model-fitting methods based on precision, proficiency, and reliability. Model-free methods need at least three heating rate runs in order to reckon the kinetic parameters without any assumptions for the reaction mechanism (Nawaz and Kumar (2022) [11]). In this section, four types of isoconversional methods have been performed to calculate the activation energy *E_a_* in kJ mol^−1^. The fitted linear equations of FR, FWO, KAS, and STK at conversions ranging from 0.1 to 0.9 for the five sets are shown in Figure 3, and the summaries of the calculated values of the activation energies are presented in Figure 4 and Table 6, respectively. Generally, model-free methods are more confident than model-fitting methods because their activation energy values are consistent with the non-isothermal data. This also helps to highlight the complexity of multi-reaction since the activation energy calculation relates to the conversion (Vyazovkin and Wight (1999) [18]). Table 6 shows the *E_a_* values for the four methods (FR, FWO, KAS, and STK) for the (0.1–0.9) conversion ranges. Due to some approximations and mathematical simplification, the results slightly differ from each other with this range of conversions. The average values are 141, 112, 106, and 108 kJ mol^−1^ for FR, FWO, KAS, and STK, respectively. These values are in good agreement with the published paper by Xu et al. (2018) [19]. Aboulkas et al. (2010) [7] calculated the activation energy of the pyrolytic reaction of PP from non-isothermal TGA data. They obtained the values of the activation energy by FR, KAS, and FWO between 179 and 188 kJ mol^−1^. Singh et al. (2021) [20] supported this finding by obtaining the activation energy of an individual corn cob and PE pyrolysis by FR, which was higher than that of KAS, FWO, and STK. In addition, Figure 4 shows that all the values of the activation energies calculated by the FR method (differential method) are slightly higher than those of the rest of the methods (FWO, KAS, and STK) (integral method), and this finding is in agreement with Aboulkas’s finding. Aboulkas et al. (2010) [7] reported that the FR method is more sensitive than the rest at low and high conversions. As expected, FR, which is the only differential method using the point value of the overall reaction rate, will produce an activation energy value that is different from the integral method using the history of the system (Aboulkas et al. (2010) [7], Mishra et al. (2015) [21], Naqvi et al. (2018) [22]). They attributed this difference to the systemic error between FWO, KAS, and STK. Naqvi et al. (2018) [22] attributed this difference to the variation in the non-linearity. Janković and Manić (2021) [23] attributed the difference between them to the consequence in the calculation formula, the equations origin, and the final results. Table 7 shows the activation energy values for PP from different published papers. It can be concluded that the activation energy values of this paper are between the highest value (225 kJ mol^−1^) by FWO and the lowest value (67 kJ mol^−^) by FR (Paik and Kar (2008) [24]).

### 3.3. Model-Fitting Kinetics Parameters Calculation

The first CR has been used by plotting $\ln\left[\frac{g\left(\alpha \right)}{T^{2}}\right]$ versus 1/T with a set of formulas of functions $g\left(\alpha \right)$ to represent the solid-state reaction. This plot produces only one reaction straight line, and the values of the activation energy obtained by the CR method with different functions $g\left(\alpha \right)$ (F1–P4) are shown in Table 8. A big difference can be seen in the value of *E_a_* depending on the model reaction mechanism (F1–P4). This abnormal value (between the lowest 14 kJ mol^−1^ and highest 380 kJ min^−1^) can be attributed to the fact that the mechanisms of these values do not meet the pyrolysis of the PP polymer.

The corrected value for this activation will be decided according to the comparison between the right side and the left side of the Criado equation with the proper model mechanism reaction, as shown in Figure 5 and Table 9. The values of *E_a_* for the data A2, A3, A4, P2, P3, and P4 are not taken into consideration because their values are far from the published *E_a_* values. The graphs of D1, A1, R1, P2, P3, and P4 are deleted from Figure 5 because their curves are not within the experimental curve range. Table 9 lists the values of *E_a_*, ln (*A*_0_), and *R*^2^ for each test (five tests). It shows the most controlling model reaction mechanism: the geometrical contraction models—contracting cylinder (R3). It has been noticed that some researchers use either CR only by assuming some mechanism model reaction, such as the Reaction Order Models—First- (F1), Second- (F2), and Third-order (F3) as a first trend or such as the current research, where it will be used with the Criado method with the free-model methods to find the appropriate mechanism model as a second trend.

In the first trend, Ali et al. (2020) [26] used the CR method with the zero- to four-order reaction order models to determine the proper order of the reaction by selecting the most suitable plot. Baloch et al. (2019) [27] considered different orders of reactions: 0, 0.5, 0.741, 1, 1.5, and 2, and the data fitted much better for 0.741. Balsora et al. (2022) [28] used the analytical method to optimize the best value of the reaction order. Bu et al. (2019) [29] presented the kinetic parameters for orders 1, 2, and 3. Lai et al. (2021) [30] and Li et al. (2022) [31] used only the first order of CR because it is considered as the main mechanism. Patidar et al. (2022) [32] revealed that the diffusion model was best suited to reflect the degradation process using the CR method.

In the second trend, Aboulkas et al. (2010) [7] determined the “Contracting Cylinder” (R3) model as an appropriate conversion model for PP. Mumbach et al. (2019) [10] chose F1 control for the third or last stage, which is mainly considered as the thermal cracking of LDPE, HDPE, and PP as the major fraction and the second PVC thermal decomposition as the minor fraction. They mentioned that the last-stage reaction includes a major decomposition of PP followed by LDPE and HDPE, with a low fraction of PVC dehychlorination. The *E* values obtained for this stage by the FWO, KAS, STK, and VYA methods were, respectively, 266.82, 268.39, 269.09, and 268.63 kJ mol^−1^. Khodaparasti et al. (2022) [33] selected the F3, F2, and F3 models for microalgae Chlorella vulgaris and municipal sewage sludge and co-pyrolysis, respectively, as the best models for predicting the solid-state mechanism. Singh et al. (2021) [20] found the mechanism reaction of the third-order reaction (F3), diffusion Jander (D3), and Ginstling–Brounshtein (D4) models to be best suited for CC pyrolysis, PE pyrolysis, and co-pyrolysis, respectively. Wan and Huang (2021) [34] used the master plot to obtain the first-order reaction model (D1), which was the most suitable mechanism function for describing the pyrolysis of nylon-6 waste.

## 4. Conclusions

The TG and DTG thermograms obtained from the TGA study showed the similar shapes and trends at different polymers’ compositions. The TGA data confirmed the existence of only one main reaction region. In this paper, two approaches have been achieved for obtaining the TGA kinetics data. In the first approach, four (FR, FWO, KAS, and STK) model-free methods are achieved, and in the second approach, two model-fitting methods, to converge the TGA data with the straight-line, have been used to calculate the activation energy.

In the first one, the average values of the activation energies of PP polymer pyrolysis at different conversions are in good agreement with the average (141, 112, 106, 108 kJ mol^−1^) for FR, FWO, KAS, and STK, respectively. 

Model-fitting (CR and Criado) methods have been applied together to obtain the activation energy and the model reaction mechanism. The values of *Ea* range from 104 to 289 kJ mol^−1^, with the geometrical contraction models—contracting cylinder (R3) as the best model reaction mechanism. However, these values are still close to the published results.

## Figures and Tables

**Figure 1 materials-16-00584-f001:**
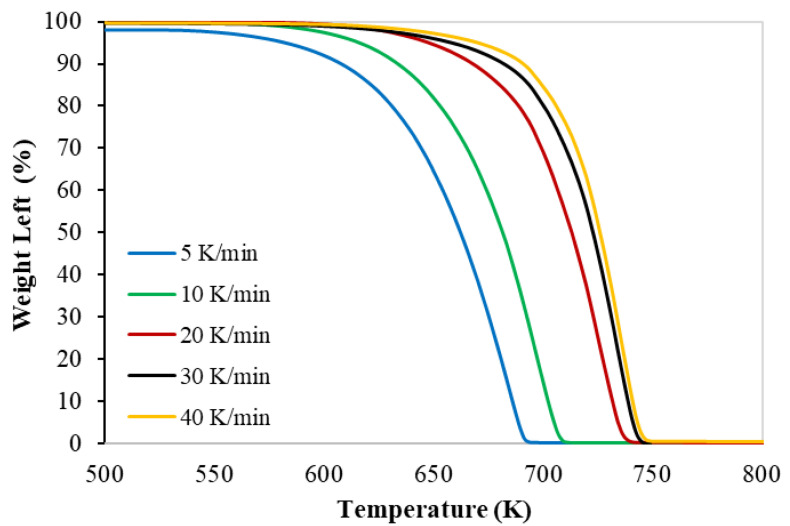
Thermogravimetric (TG) curves of the PP pyrolysis with different heating rates.

**Figure 2 materials-16-00584-f002:**
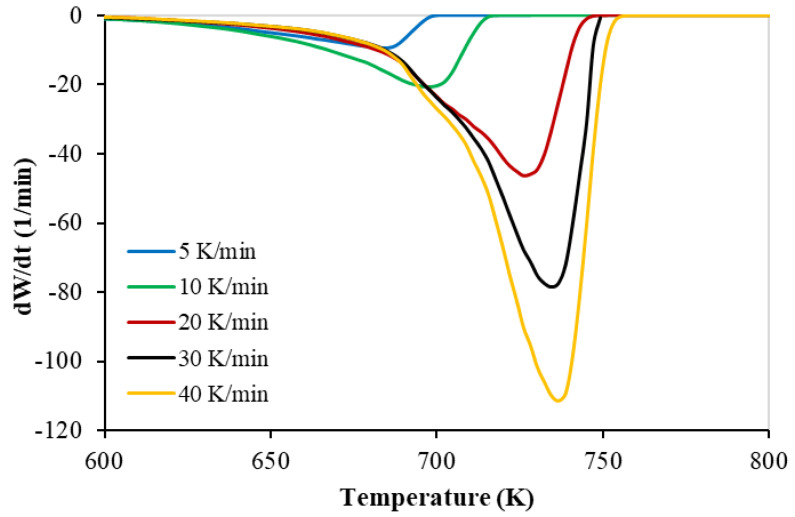
Derivative thermogravimetric (DTG) curves of the PP pyrolysis with different heating rates.

**Figure 3 materials-16-00584-f003:**
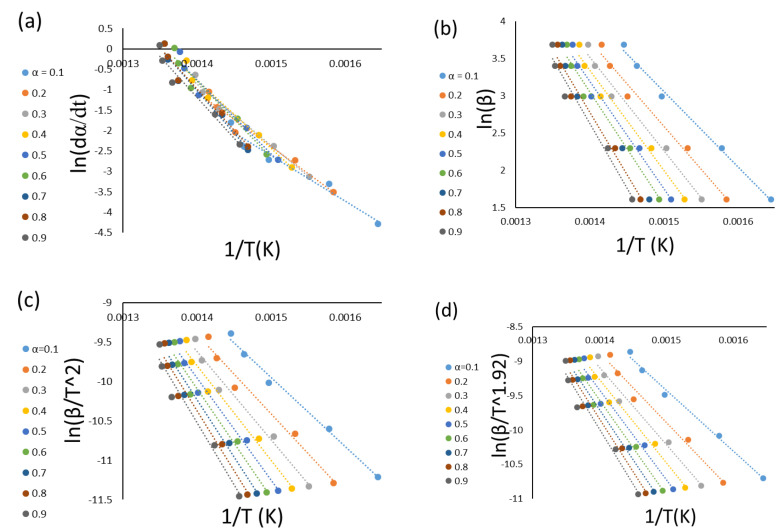
Model-free methods of the PP pyrolysis by the: (**a**) FR, (**b**) FWO, (**c**) KAS, and (**d**) STK models.

**Figure 4 materials-16-00584-f004:**
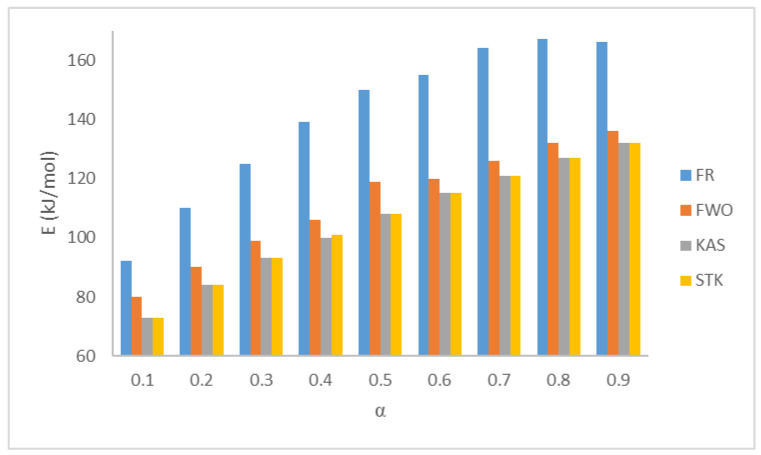
Activation energies for the FR, FWO, KAS, and STK of PP pyrolysis.

**Figure 5 materials-16-00584-f005:**
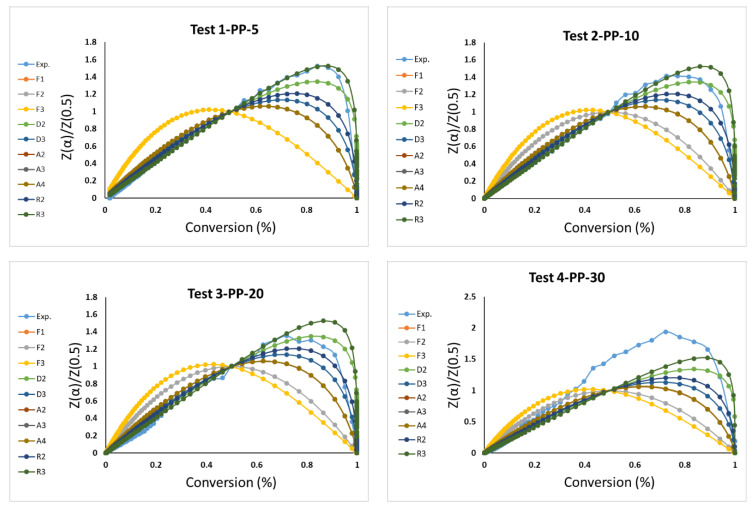
Masterplots of different kinetic models and experimental data of five tests.

**Table 1 materials-16-00584-t001:** Ultimate and proximate analysis of the polypropylene sample.

Proximate Analysis, wt%.			Ultimate Analysis, wt%			
Moisture	Volatile	Ash	C	H	N	S
0.01	99.63	0.29	85.00	14.73	0.04	0.23

**Table 2 materials-16-00584-t002:** Equations for four model-free methods.

Method	Equation	Integral (I) or Differential (D)	Plot
Friedman	$\ln\left(\beta \frac{d\alpha }{dT}\right)=\ln[A_{o}f\left(\alpha \right)]-\frac{E}{RT}$	D	$\ln\left(\beta \frac{d\alpha }{dT}\right)\;\;\mathrm{vs}.\frac{1}{T}$
Flynn–Wall–Qzawa (FWO)	$\ln\left(\beta \right)=\ln\frac{A_{o}E}{Rg\left(\alpha \right)}-5.331-1.052\frac{E}{RT}$	I	$\ln\left(\beta \right)\;\mathrm{vs}.\frac{1}{T}$
Kissinger–Akahira–Sunose (KAS)	$\ln\left(\frac{\beta }{T^{2}}\right)=\ln\frac{A_{o}R}{Eg\left(\alpha \right)}-\frac{E}{RT}$	I	$\ln\left(\beta /T^{2}\right)\;\mathrm{vs}.\frac{1}{T}$
Starink	$\ln\frac{\beta }{T^{1.92}}=-1.0008\frac{E}{RT}+C$	I	$\ln\left(\beta /T^{1.92}\right)\;\mathrm{vs}.\frac{1}{T}$

**Table 3 materials-16-00584-t003:** Equations for two model-fitting methods.

	Equation
CR method	$\ln\left[\frac{g\left(\alpha \right)}{T^{2}}\right]=\ln\left[\frac{A_{o}R}{\beta E}\right]-\frac{E}{RT}\;$
Criado method	$\frac{Z\left(\alpha \right)}{Z\left(0.5\right)}=\frac{f\left(\alpha \right)g\left(\alpha \right)}{f\left(0.5\right)g\left(0.5\right)}={\left(\frac{T_{\alpha }}{T_{0.5}}\right)}^{2}\;\frac{{\left(\frac{d\alpha }{dt}\right)}_{\alpha }}{{\left(\frac{d\alpha }{dt}\right)}_{0.5}}\;$

**Table 4 materials-16-00584-t004:** Solid-state reaction mechanism (Mumbach et al. (2019) [10]).

Reaction Mechanism	Code	*f*(*α*)	g(*α*)
Reaction order models—First order	F1	1−α	$-\ln\left(1-\alpha \right)$
Reaction order models—Second order	F2	${\left(1-\alpha \right)}^{2}$	${\left(1-\alpha \right)}^{-1}-1$
Reaction order models—Third order	F3	${\left(1-\alpha \right)}^{3}$	$[{\left(1-\alpha \right)}^{-1}-1]/2$
Diffusion model—One-dimension diffusion	D1	$1/2\alpha^{-1}$	$\alpha^{2}$
Diffusion model—Two-dimension diffusion	D2	${\left[-\ln\left(1-\alpha \right)\right]}^{-1}$	$\left(1-\alpha \right)\ln\left(1-\alpha \right)+\alpha$
Diffusion model—Three-dimension diffusion	D3	$3/2{\left[1-{\left(1-\alpha \right)}^{1/3}\right]}^{-1}$	${\left[1-{\left(1-\alpha \right)}^{1/3}\right]}^{2}$
Nucleation models—Two-dimension nucleation	A2	$2\left(1-\alpha \right){\left[-\ln\left(1-\alpha \right)\right]}^{1/2}$	${\left[-\ln\left(1-\alpha \right)\right]}^{1/2}$
Nucleation models—Three-dimension nucleation	A3	$3\left(1-\alpha \right){\left[-\ln\left(1-\alpha \right)\right]}^{1/3}$	${\left[-\ln\left(1-\alpha \right)\right]}^{1/3}$
Nucleation models—four-dimension nucleation	A4	$4\left(1-\alpha \right){\left[-\ln\left(1-\alpha \right)\right]}^{1/4}$	${\left[-\ln\left(1-\alpha \right)\right]}^{1/4}$
Geometrical contraction models—One-dimension	R1	1	α
Geometrical contraction models—Contracting sphere	R2	$2{\left(1-\alpha \right)}^{1/2}$	1−${\left(1-\alpha \right)}^{1/2}$
Geometrical contraction models—Contracting cylinder	R3	$3{\left(1-\alpha \right)}^{1/3}$	1$-{\left(1-\alpha \right)}^{1/3}$
Nucleation models—Power law	P2	$2\alpha^{1/2}$	$\alpha^{1/2}$
Nucleation models—Power law	P3	$3\alpha^{2/3}$	$\alpha^{1/3}$
Nucleation models—Power law	P4	$4\alpha^{3/4}$	$\alpha^{1/4}$

**Table 5 materials-16-00584-t005:** Onset, peak, and final temperatures for five tests.

Test No.	Heating Rate (K min^−1^)	Onset Temp. (K)	Peak Temp. (K)	Final Temp. (K)
1	5	625	687	694
2	10	630	700	713
3	20	650	725	740
4	30	663	734	747
5	40	670	737	750

**Table 6 materials-16-00584-t006:** Activation energy of PP pyrolysis calculated by four model-free methods.

Conversion	FR		FWO		KAS		STK	
	*E_a_* (kJ mol^−1^)	*R* ^2^	*E*_*a*_ (kJ mol^−1^)	*R* ^2^	*E**_a_* (kJ mol^−1^)	*R* ^2^	*E**_a_* (kJ mol^−1^)	*R* ^2^
0.1	92	0.9634	80	0.9937	73	0.9920	71	0.9921
0.2	110	0.9637	90	0.9829	84	0.9784	84	0.9786
0.3	125	0.9818	99	0.9825	93	0.9783	109	0.9785
0.4	139	0.9739	106	0.9809	100	0.9765	101	0.9767
0.5	150	0.9573	119	0.9775	108	0.9726	108	0.9729
0.6	155	0.9597	120	0.9747	115	0.9695	115	0.9697
0.7	164	0.9671	126	0.9723	121	0.9669	121	0.9671
0.8	167	0.9643	132	0.9703	127	0.9647	127	0.9650
0.9	166	0.9535	136	0.9676	132	0.9618	132	0.9621
**Average**	**141**	**0.9650**	**112**	**0.9780**	**106**	**0.9734**	**108**	**0.9736**

**Table 7 materials-16-00584-t007:** Activation energies of PP obtained by different published papers.

References	E (kJ mol^−1^)	Method
Aboulkas et al. (2010) [7]	179–183	FR, FWO, KAS
Wu et al. (1993) [4]	184	FR
Paik and Kar (2008) [24]	67–241 125–224	FR FWO
Galiwango and Gabbar (2022) [11]	165.54 159.72 187.25	FWO FR CR
Mortezaeikia et al. (2021) [25]	261.22 196–214	KAS KAS

**Table 8 materials-16-00584-t008:** Activation energies of the PP polymer by the CR method.

Reaction Mechanism	Code	Test 1 PP-5			Test 2 PP-10			Test 3 PP-20		
		*E*_*a*_ (kJ mol^−1^)	ln (*A*_0_)	*R* ^2^	*E*_*a*_ (kJ mol^−1^)	ln (*A*_0_)	*R* ^2^	*E*_*a*_ (kJ mol^−1^)	ln (*A*_0_)	*R* ^2^
Reaction order models—First-order	F1	125	20.51	0.9994	143	23.85	0.9994	179	29.59	0.9988
Reaction order models—Second-order	F2	168	29.13	0.9976	186	32.07	0.9972	214	35.93	0.9973
Reaction order models—Third-order	F3	220	39.16	0.9953	236	41.5	0.9945	252	42.94	0.9955
Diffusion models—One-dimension	D1	189	31.5	0.9999	225	37.76	0.9999	307	50.87	0.9996
Diffusion models—Two-dimension	D2	210	35	1.0000	247	41.21	1	327	53.68	0.9995
Diffusion models—Three-dimension	D3	235	38.32	0.9998	271	44.38	0.9998	348	55.96	0.9992
Nucleation models—Two-dimension	A2	57	13.29	0.9993	66	12.9	0.9993	84	12.98	0.9982
Nucleation models—Three-dimension	A3	34	16.82	0.9992	40	16.87	0.9992	52	16.26	0.9985
Nucleation models—Fourth-dimension	A4	23	18.45	0.999	27	18.71	0.9993	36	18.54	0.9987
Geometrical contraction models—One-dimension phase boundary	R1	89	13.33	0.9999	107	16.88	0.9999	148	23.91	0.9996
Geometrical contraction models—Contracting sphere	R2	106	16.05	0.9999	124	19.52	0.9999	163	25.97	0.9993
Geometrical contraction models—Contracting cylinder	R3	112	16.86	0.9998	130	20.29	0.9998	168	26.52	0.9992
Nucleation models—Power law	P2	39	16.33	0.9999	48	15.92	0.9999	68	13.99	0.9996
Nucleation models—Power law	P3	22	18.66	0.9998	28	18.74	0.9999	42	17.88	0.9995
Nucleation models—Power law	P4	14	19.66	0.9997	18	19.97	0.9998	28	19.66	0.9994
**Reaction Mechanism**	**Code**	**Test 4 PP-30**			**Test 5 PP-40**					
		** *E* _ *a* _ ** **(kJ mol^−1^)**	**ln (*A*_0_)**	** *R* ^2^ **	** *E* _ *a* _ ** **(kJ mol^−1^)**	**ln (*A*_0_)**	** *R* ^2^ **			
Reaction order models—First-order	F1	198	32.87	0.9989	212	35.29	0.9984			
Reaction order models—Second-order	F2	245	41.15	0.9972	251	42.14	0.9969			
Reaction order models—Third-order	F3	297	50.45	0.9953	293	49.67	0.9952			
Diffusion models—One-dimension	D1	328	53.87	0.9999	367	60.477	0.9995			
Diffusion models—Two-dimension	D2	352	57.59	0.9997	388	63.6	0.9993			
Diffusion models—Three-dimension	D3	380	60.95	0.9993	412	66.21	0.9989			
Nucleation models—Two-dimension	A2	93	14.85	0.9987	100	16.24	0.9983			
Nucleation models—Three-dimension	A3	58	15.88	0.9986	63	15.56	0.9981			
Nucleation models—Fourth-dimension	A4	41	18.41	0.9984	44	15.2	0.9978			
Geometrical contraction models—One-dimension phase boundary	R1	158	25.63	0.9998	177	29.13	0.9995			
Geometrical contraction models—Contracting sphere	R2	177	28.43	0.9995	194	31.43	0.9991			
Geometrical contraction models—Contracting cylinder	R3	184	29.24	0.9993	200	32.06	0.9989			
Nucleation models—Power law	P2	73	13.86	0.9998	83	13.07	0.9995			
Nucleation models—Power law	P3	45	17.97	0.9998	51	17.34	0.9994			
Nucleation models—Power law	P4	31	19.89	0.9998	35	19.52	0.9993			

**Table 9 materials-16-00584-t009:** Activation energy of mixed polymers pyrolysis by model-fitting methods: CR.

Test No.	*E_a_* (kJ/mol)	ln (*A*_0_)	*R* ^2^	Reaction Mechanism
1	112	16.86	0.9998	Geometrical contraction models—Contracting cylinder (R3)
2	130	20.29	0.9998	Geometrical contraction models—Contracting cylinder (R3)
3	168	26.52	0.9993	Geometrical contraction models—Contracting cylinder (R3)
4	184	29.24	0.9993	Geometrical contraction models—Contracting cylinder (R3)
5	200	32.06	0.9989	Geometrical contraction models—Contracting cylinder (R3)

## Data Availability

Not applicable.

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
