# Peer review of "Kinetics Study of Polypropylene Pyrolysis by Non-Isothermal Thermogravimetric Analysis"

_materials, 2023, doi:10.3390/ma16020584_

Round 1
Reviewer 1 Report
As a general-purpose plastic, the application of polypropylene is increasing day by day, so it is important to study the thermal decomposition properties of polypropylene. Although the thermal decomposition behavior of polypropylene has been studied for decades, this paper still adopts different methods to deal with TGA data of polypropylene in a relatively comprehensive and systematic way and obtains a series of decomposition activation energy data based on different models.
The reviewers believe that the revision is ready for publication. The authors need to add some minor issues.
1. the relative molecular mass and specific type of samples should be given.
2. When comparing with previous decomposition activation energy data, the differences in molecular weight, molecular weight distribution, sequence structure and other properties should be considered.
Author Response
Response to the Comments of Reviewer 1
Comments and Suggestions for Author
As a general-purpose plastic, the application of polypropylene is increasing day by day, so it is important to study the thermal decomposition properties of polypropylene. Although the thermal decomposition behavior of polypropylene has been studied for decades, this paper still adopts different methods to deal with TGA data of polypropylene in a relatively comprehensive and systematic way and obtains a series of decomposition activation energy data based on different models.
The reviewers believe that the revision is ready for publication. The authors need to add some minor issues.
1. the relative molecular mass and specific type of samples should be given.
Comment: Thank you for this comment
Action: I put all the information about the polymer in the paper and according to the standard requirement for any paper published before. I will be happy to add these information if it is available.
When comparing with previous decomposition activation energy data, the differences in molecular weight, molecular weight distribution, sequence structure and other properties should be considered.
Comment: Thank you for this comment
Action: Again referring to the previous comments, I could not mention the effect of these parameters because our TGA is not connected to any instrument like GC to monitor the distribution of the products. This issue is used to discuss if you run laboratory scale batch reactor or connect TG or MS to the TGA. The only reaction involving in the pyrolysis of PP by TGA is the decomposition of high molecular weight into smaller molecular weight ones.
I am attaching some papers highlighting this issue:
For more information on the pyrolysis of different polymers can be found in the following paper:
Many thanks for your kind understanding.
Thanks again with best regards,
Dr. Ibrahim Dubdub

Reviewer 2 Report
Plastic waste pyrolysis is a very up-to-date topic. My main concern is that this work does not appear to be novel. The conclusion states that "these values are still very close to the published results." If these results were already published, why bother to re-do them? This has to be very well explained by the authors if this manuscript is considered for publication.
Below please see my additional comments:
Abstract
- PP is one of the main representatives of plastic wastes, not the main
Introduction
- page 1, first paragraph - more information is necessary, why only compare to incineration? what about hydrothermal processing (HTP), etc.?
- there are more papers focusing on PP with reaction pathways (i.e., papers discussing HTP or pyrolysis reaction pathways)
Materials and Methos
- why were the PP pellets ground into powder?
- what was the powder size? Would this be possible in an actual application?
- Table 1 - where are the N and S coming from? Colors?
- page 5, lines 146-147 - that was already said on the previous page
Author Response
Response to the Comments of Reviewer 2
Comments and Suggestions for Author
Plastic waste pyrolysis is a very up-to-date topic. My main concern is that this work does not appear to be novel. The conclusion states that "these values are still very close to the published results." If these results were already published, why bother to re-do them? This has to be very well explained by the authors if this manuscript is considered for publication.
Below please see my additional comments:
1- Abstract
- PP is one of the main representatives of plastic wastes, not the main
Comment: Thank you for this comment
Action: PP is one of the six main polymers for Plastic wastes, therefore, it can be rephrased the sentence to be (Line: 6):
“Polypropylene (PP) is considered as a one of six polymers representative of plastic wastes.”
Action: more details can be found in the following references:
2- Introduction
- page 1, first paragraph - more information is necessary, why only compare to incineration? what about hydrothermal processing (HTP), etc.?
Comment: Thank you for this comment
Action: I highlighted these topics in different way in my publication:
Therefore, hydrothermal processing is considered one of the advanced technique “tertiary recycling”
- there are more papers focusing on PP with reaction pathways (i.e., papers discussing HTP or pyrolysis reaction pathways)
Comment: Thank you for this comment
Action: I will be very if you provide me any paper with the reaction pathways (pyrolysis or HTP) for TGA instrument. TGA cannot release or give more information especially when there is only one reaction within the temperature range.
Materials and Methos
- why were the PP pellets ground into powder?
Comment: Thank you for this comment
Action: Because TGA needs very small amount (10-30 mg), therefore, the PP pellets should be grounded to powder before feeding to the TGA in order to increase the distribution of the heat thorough the sample.
- what was the powder size? Would this be possible in an actual application?
Comment: Thank you for this comment
Action: Depending on the type of TGA instrument, the powder size is ranging from small value <750 μm (micro meter) to large value 1mm (milli meter). For our experiment, the powder size is about 10 mm
- Table 1 - where are the N and S coming from? Colors?
Comment: Thank you for this comment
Action: These elements analysis has been obtained from (mentioned in other paper):
“The proximate and ultimate analysis of PP were conducted using PerkinElmer Simultaneous Thermal Analyzer (STA) 6000, and PerkinElmer 2400 Series II CHNS Elemental Analyzer, Waltham, MA, USA, respectively.”
- page 5, lines 146-147 - that was already said on the previous page
Comment: Thank you for this comment
Action: sorry for this duplication. It has been deleted.
Many thanks for your kind understanding.
Thanks again with best regards,
Dr. Ibrahim Dubdub
